# Molecular mechanisms of human P2X3 receptor channel activation and modulation by divalent cation bound ATP

Mufeng Li[1], Yao Wang[2], Rahul Banerjee[3], Fabrizio Marinelli[3], Shai Silberberg[1], José D Faraldo-Gómez[3]*, Motoyuki Hattori[2]*, Kenton Jon Swartz[1]*

[1]Molecular Physiology and Biophysics Section, Porter Neuroscience Research Center, National Institute of Neurological Disorders and Stroke, National Institutes of Health, Bethesda, United States; [2]State Key Laboratory of Genetic Engineering, Collaborative Innovation Center of Genetics and Development, Multiscale Research Institute for Complex Systems, Department of Physiology and Biophysics, School of Life Sciences, Fudan University, Shanghai, China; [3]Theoretical Molecular Biophysics Laboratory, National Heart, Lung and Blood Institute, National Institutes of Health, Bethesda, United States

**Abstract** P2X3 receptor channels expressed in sensory neurons are activated by extracellular ATP and serve important roles in nociception and sensory hypersensitization, making them attractive therapeutic targets. Although several P2X3 structures are known, it is unclear how physiologically abundant $Ca^{2+}$-ATP and $Mg^{2+}$-ATP activate the receptor, or how divalent cations regulate channel function. We used structural, computational and functional approaches to show that a crucial acidic chamber near the nucleotide-binding pocket in human P2X3 receptors accommodates divalent ions in two distinct modes in the absence and presence of nucleotide. The unusual engagement between the receptor, divalent ion and the γ-phosphate of ATP enables channel activation by ATP-divalent complex, cooperatively stabilizes the nucleotide on the receptor to slow ATP unbinding and recovery from desensitization, a key mechanism for limiting channel activity. These findings reveal how P2X3 receptors recognize and are activated by divalent-bound ATP, aiding future physiological investigations and drug development.
DOI: https://doi.org/10.7554/eLife.47060.001

*For correspondence:
jose.faraldo@nih.gov (JéDF-Gó);
hattorim@fudan.edu.cn (MH);
swartzk@ninds.nih.gov (KJS)

## Introduction

P2X3 receptors are a family of trimeric cation channels that are activated by extracellular ATP. They play particularly important roles in sensory neurons where their activation is critical for gustatory and nociceptive responses (*Barclay et al., 2002*; *Chen et al., 1995*; *Cook et al., 1997*; *Finger et al., 2005*; *Huang et al., 2011*; *North, 2003*; *North, 2004*; *Vandenbeuch et al., 2015*), visceral reflexes and sensory hypersensitization (*Cockayne et al., 2000*; *Deiteren et al., 2015*; *Ford et al., 2015*; *Souslova et al., 2000*). Accordingly, P2X3 knock-out or knock-down in rodents reduces nociception and visceral reflexes (*Barclay et al., 2002*; *Cockayne et al., 2000*; *Souslova et al., 2000*; *Vlaskovska et al., 2001*), and the P2X3 receptor antagonist A-317491 reduces inflammatory and neuropathic pain (*Hansen et al., 2012*; *Jarvis et al., 2002*). Clinical studies in humans have shown that high-affinity P2X3 antagonist AF-219 is well tolerated and can significantly reduce cough frequency in patients with refractory chronic cough (*Abdulqawi et al., 2015*; *Ford and Undem, 2013*; *Wang et al., 2018*). P2X3 receptor antagonists are also being developed for migraine, itch and cancer pain (*Ginnetti et al., 2018*; *North, 2003*).

P2X3 homomeric receptors activate and desensitize on the millisecond timescale upon binding ATP but unbind ATP and recover from desensitization on the timescale of minutes (*Cook and McCleskey, 1997*; *Giniatullin and Nistri, 2013*; *Sokolova et al., 2006*). This unusually long recovery time limits channel activation frequency, which may be an important protective mechanism for preventing sensory hypersensitivity. The molecular mechanisms underlying these unique gating processes remain to be fully established. X-ray structures of human P2X3 (hP2X3) receptor channels have recently been solved in the absence of ATP, as well as bound to free-ATP (i.e. ATP not complexed with a divalent ion) or to antagonists (*Mansoor et al., 2016*). These structures elegantly reveal how free-ATP binds to the large extracellular domain to open the transmembrane pore, and to subsequently close during the process of desensitization. Specifically, the nucleotide adopts a bent pose with the adenosine ring positioned above an aromatic residue (F174), the $\alpha$−phosphate group interacting with S275, and the $\beta$− and $\gamma$−phosphates intimately engaging with K63, K65, K299, R281 and N279 (*Figure 1D*) (*Hattori and Gouaux, 2012*; *Mansoor et al., 2016*).

Although free-ATP is an agonist in laboratory conditions, P2X3 receptors are activated primarily by $Ca^{2+}$-ATP and $Mg^{2+}$-ATP under physiological conditions because ATP released from cells at micromolar concentrations would largely be in complex with $Ca^{2+}$ and $Mg^{2+}$, both of which are present in millimolar concentrations in extracellular physiological solutions (*Lazarowski et al., 2003*;

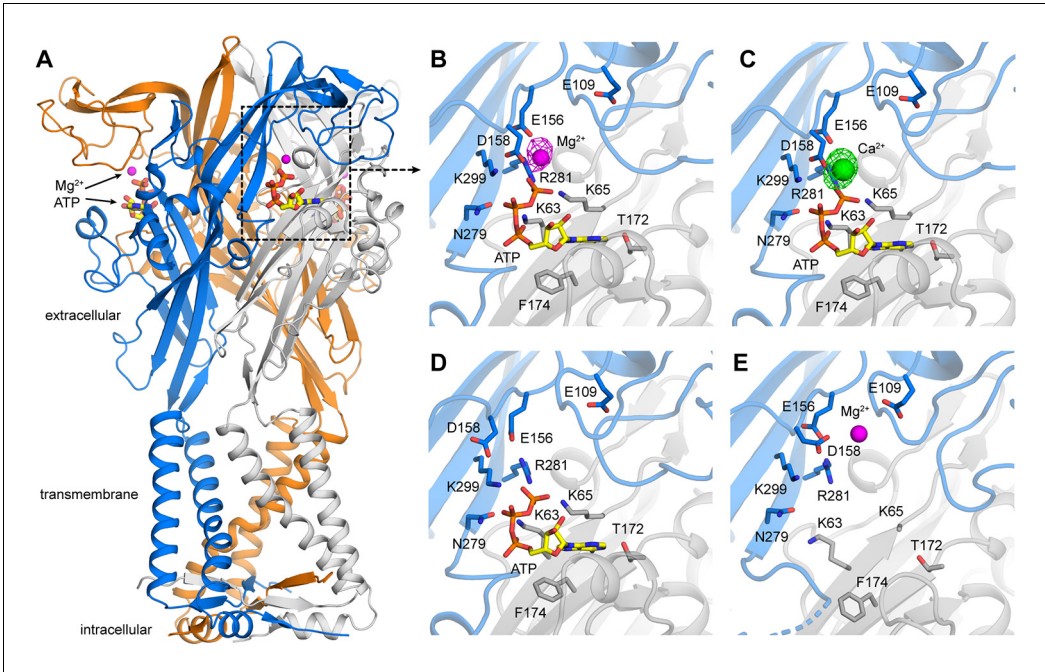

**Figure 1.** X-ray structures of hP2X3 receptors with divalent ions bound.. (A) Overall structure of hP2X3 MFC$_{slow}$ with ATP and $Mg^{2+}$ bound (PDB ID 6AH5). Agonist binding pockets are formed at each of the three interfaces between adjacent subunits (blue, orange, gray). (B) Close-up view of the binding site for $Mg^{2+}$-ATP in hP2X3 MFC$_{slow}$, highlighting key interactions between the receptor and both ligands. In this 'lower mode', $Mg^{2+}$ interacts directly with the $\gamma$ phosphate of ATP and acidic residues in the receptor. The omit *Fo-Fc* density map for $Mg^{2+}$ is overlaid (magenta mesh), contoured at 3.5$\sigma$. (C) Same as B), for the structure of hP2X3 MFC$_{slow}$ with ATP and $Ca^{2+}$ bound (PDB ID 6AH4). The omit *Fo-Fc* density map for $Ca^{2+}$ (green) is contoured at 5$\sigma$. (D) Same view of the agonist-binding pocket in an existing structure of hP2X3 MFC$_{slow}$ bound to ATP only (PDB ID 5SVK). Note the mode of ATP binding is nearly identical to that in B) and C), except for the acidic sidechains near $Mg^{2+}$ or $Ca^{2+}$. (E) Same view of the agonist-binding pocket in an existing structure of hP2X3 MFC$_{slow}$ bound to $Mg^{2+}$ only (PDB ID: 5SVJ). In this 'upper mode', $Mg^{2+}$ shifts upwards and is stabilized by a cluster of acidic sidechains. .

DOI: https://doi.org/10.7554/eLife.47060.002

The following figure supplement is available for figure 1:

**Figure supplement 1.** Conformational changes associated with ATP binding in the presence and absence of $Mg^{2+}$.

DOI: https://doi.org/10.7554/eLife.47060.003

*Li et al., 2013*). From the available structure of hP2X3 bound by free-ATP, it is unclear how the receptor might accommodate a divalent-bound nucleotide. When ATP is in solution, the divalent ion preferably interacts with the β− and γ−phosphate groups simultaneously, or alternatively, with all three phosphate groups (*Branduardi et al., 2016*; *Cohn and Hughes, 1962*). Whether these interactions are compatible with those observed between free ATP and the receptor in the ATP-bound structure is unclear; even if they are, the presence of divalent ions would effectively weaken the interaction between agonist and receptor, which would be physiologically counterintuitive. It would seem reasonable to anticipate that when bound to the receptor, $Mg^{2+}$ and ATP interact differently than when in solution. Intriguingly, a $Mg^{2+}$ binding site was identified in previous structures of hP2X3 in the absence of ATP (apo), or in the presence of antagonists (*Figure 1E*) (*Mansoor et al., 2016*; *Wang et al., 2018*). However, from the comparison of free-ATP bound and $Mg^{2+}$ bound apo structures, this site would be too far from the nucleotide-binding pocket to directly interact with ATP (*Figure 1D,E*), making its functional significance unclear. To add to the complexity of P2X3 receptor activation in the presence of divalent cations, $Mg^{2+}$ has been reported to inhibit P2X3 receptors and $Ca^{2+}$ to stimulate the activity of the channel (*Cook et al., 1998*; *Giniatullin et al., 2003*; *Li et al., 2013*). To elucidate the mechanisms of these physiologically and therapeutically important ion channels, it is essential to determine the sites and mechanisms of divalent ion action. We therefore set out to elucidate the mechanism by which $Ca^{2+}$-ATP and $Mg^{2+}$-ATP activate P2X3 receptors, and to explore how divalent cations regulate the receptor.

## Results

### Structures of hP2X3 receptor reveal $Mg^{2+}$-ATP and $Ca^{2+}$-ATP bind in a unique mode

We began by crystallizing the hP2X3 MFC$_{slow}$ construct (*Mansoor et al., 2016*) in solutions containing ATP with either $Mg^{2+}$ or $Ca^{2+}$. This construct shows minimal desensitization (*Hausmann et al., 2014*; *Mansoor et al., 2016*), and was previously used to obtain an open state structure of hP2X3 with free-ATPbound. We solved X-ray structures of the receptor with $Mg^{2+}$-ATP or $Ca^{2+}$-ATP bound at resolutions of 3.8 Å and 3.3 Å, respectively (*Supplementary file 1*). The pores appear to be in an open-like conformation in both structures, similar to that observed in the structure of hP2X3 bound to free-ATP (*Figure 1—figure supplement 1*). Importantly, we observed strong residual electron densities around the ATP binding site in these structures (*Figure 1B,C*). Considering that the crystallization conditions included either 50 mM Mg-acetate or 50 mM Ca-acetate with no other divalent cations, and that the electron densities are surrounded by acidic residues, we interpreted the densities as $Mg^{2+}$ and $Ca^{2+}$ ions, respectively. In both structures, the divalent ions are detected at the edge of the ATP binding pocket, simultaneously interacting with D158 on the receptor and with the γ-phosphate of ATP; E156 is also positioned nearby and appears to be involved in coordinating the ion. Because the similar binding sites we observed for $Ca^{2+}$ and $Mg^{2+}$ imply a common mechanism for activation by $Ca^{2+}$-ATP and $Mg^{2+}$-ATP, we largely focused our efforts on $Mg^{2+}$-ATP so we could take advantage of the more extensive data on this form of divalent-bound ATP.

A survey of the Protein Data Bank (PDB) reveals that the geometry of the $Mg^{2+}$-ATP and $Ca^{2+}$-ATP complexes observed in our structures is highly unusual (*Figure 2A*). Among non-redundant structures determined at 2.5 Å resolution or better with $Mg^{2+}$-ATP (or $Ca^{2+}$-ATP) bound (see Materials and methods), only 4% (or none) feature a complex in which $Mg^{2+}$ is coordinated solely by the γ-phosphate (and 2% for the AMP-PNP analog). The prevalent geometries are instead those in which $Mg^{2+}$ is coordinated by both the β- and γ-phosphates or by all three phosphate groups concurrently. These two interaction modes are also the most energetically favorable when $Mg^{2+}$-ATP is in solution (*Branduardi et al., 2016*). That the $Mg^{2+}$-ATP complex observed in our structures adopts a rare configuration suggests that interactions between the receptor and divalent ions have a dominant effect in determining the structure and stability of the complex.

The ion binding mode we observe with ATP bound, which we designate as the 'lower' mode, is distinct from that observed previously in structures without ATP. In those previous structures, $Mg^{2+}$ was proposed to be coordinated by the side chains of E109 and D158, and the carbonyl oxygen of E156, which we designate as the 'upper' mode (*Figure 1E*) (*Mansoor et al., 2016*). Although $Mg^{2+}$ in the upper mode is near the ATP binding pocket, it is too far to interact directly with the bound

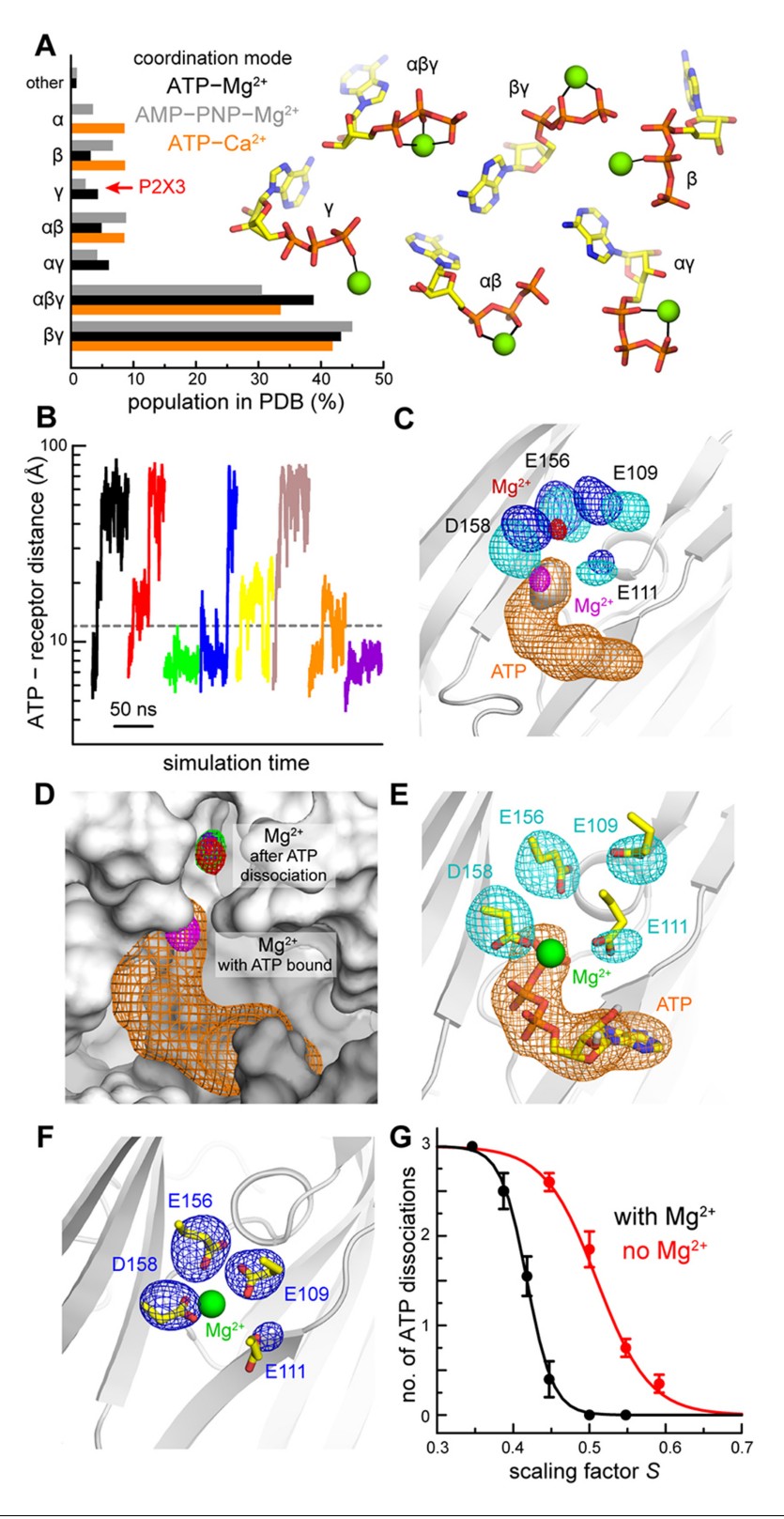

**Figure 2.** Computational study of the mode of ATP and $Mg^{2+}$ recognition by P2X3. (**A**) Occurrence of different modes of $Mg^{2+}$-ATP and $Ca^{2+}$-ATP interaction in non-redundant high-resolution crystal structures available in the Protein Data Bank. Each mode is indicated by the phosphate groups in ATP that coordinate the cation. Analogous data is shown for complexes of $Mg^{2+}$ with the non-hydrolyzable analog AMP-PNP. (**B**) A sample of the simulated trajectories of the P2X3-ATP-$Mg^{2+}$ complex produced with the solute-tempering protocol ($S$ = 0.39). Trajectories showing spontaneous ATP

*Figure 2 continued on next page*

*Figure 2 continued*

dissociation from P2X3 (and $Mg^{2+}$) are those in which the distance between the center of ATP and the center of the binding site in the receptor exceed 12 Å. Additional trajectories are depicted in *Figure 2—figure supplement 1*. (C) Comparison of P2X3-ATP-$Mg^{2+}$ and P2X3-$Mg^{2+}$ following ATP dissociation. For P2X3-ATP-$Mg^{2+}$, 3D density maps calculated from the simulation data (see Materials and methods) are shown for ATP (orange), the γ-phosphate (gray surface), $Mg^{2+}$ (magenta) and the surrounding carboxyl groups of acidic residues (cyan). Analogous maps are shown for ion and sidechains (red and blue, respectively), following ATP dissociation. The maps shown are averages of the data for the three protein subunits. The contour levels used for $Mg^{2+}$ are the same in both cases; likewise for the sidechains. The protein conformation (gray cartoons) is a snapshot of the ATP-bound state. (D) Same as panel C, for $Mg^{2+}$ and ATP in the ATP-bound state. For the ATP-free state, individual $Mg^{2+}$ density maps are shown for each binding site in the trimer (red, green, purple), to demonstrate the observed displacement is reproducible. The protein snapshot is that in panel C, now represented as a surface, with adjacent protein subunits in light and dark gray. (E) Same as panel C, for the ATP-bound state, now showing a randomly selected configuration of $Mg^{2+}$, ATP and the four acidic sidechains that represents the density. Non-polar hydrogen atoms and other components of the simulation system are omitted for clarity. (F) Same as panel E, for the ATP-dissociated state, in a slightly different view, for clarity. (G) Impact of $Mg^{2+}$ on the nature of the interaction between ATP and P2X3. The plot quantifies the number of ATP dissociation events expected to occur in a 50-ns time-window, deduced from our simulation data, for each value of the scaling factor $S$, and in the presence or absence of $Mg^{2+}$. Note that one ATP molecule is bound to each of the three protein subunits, and hence the maximum number of dissociation events is also 3. Each data point is an average of $N = 20$ independent simulations. Error bars denote the S.E.M. The data was fitted to a sigmoidal function $f = 3/(1 + \exp [m (S - n)])$. The resulting parameters (C.C. > 99%) are: with $Mg^{2+}$, $m = 58.6$, $n = 0.42$; without $Mg^{2+}$, $m = 29.1$, $n = 0.51$.

DOI: https://doi.org/10.7554/eLife.47060.004

The following figure supplements are available for figure 2:

**Figure supplement 1.** Simulated ATP dissociation from P2X3 in the presence and in the absence of $Mg^{2+}$.

DOI: https://doi.org/10.7554/eLife.47060.005

**Figure supplement 2.** Change in the mode of $Mg^{2+}$ binding to P2X3 following ATP dissociation.

DOI: https://doi.org/10.7554/eLife.47060.006

**Figure supplement 3.** Cooperativity between $Mg^{2+}$ and P2X3 slows down ATP dissociation.

DOI: https://doi.org/10.7554/eLife.47060.007

nucleotide. Conversely, in our ATP-bound structure, with $Mg^{2+}$ directly coordinated by the ATP γ-phosphate and D158, the ion is ~9 Å away from the side chain of E109. The key residue involved in both binding modes is D158, whose side chain points towards E109 for the upper mode or towards the γ-phosphate of ATP for the lower mode (*Figure 1B–E*). Therefore, these structures not only reveal the mode of recognition of ATP in its physiologically predominant forms with $Ca^{2+}$ or $Mg^{2+}$ bound, but also demonstrate how the acidic chamber near the nucleotide binding pocket enables the receptor to accommodate $Mg^{2+}$ in two distinct configurations.

## Simulations reveal $Mg^{2+}$ displacement to upper mode results from ATP dissociation

Our crystal structures suggest a correlation between the mode of $Mg^{2+}$ binding to P2X3, that is lower or upper mode, and the presence or absence of ATP. To evaluate whether there is a causal relationship between these two observations, we devised a molecular-simulation protocol to induce the spontaneous dissociation of ATP from the receptor and evaluated how $Mg^{2+}$ responds to this event (see Materials and methods). In this protocol, based on the solute-tempering method (*Liu et al., 2005*), a scaling factor is used to weaken the interactions of a ligand (i.e. ATP) with both the surrounding solvent and receptor (i.e. P2X3 with bound $Mg^{2+}$). As a result, the free-energy barriers and minima that underlie ligand binding and unbinding become shallower, accelerating the kinetics of these processes exponentially. Dissociation trajectories

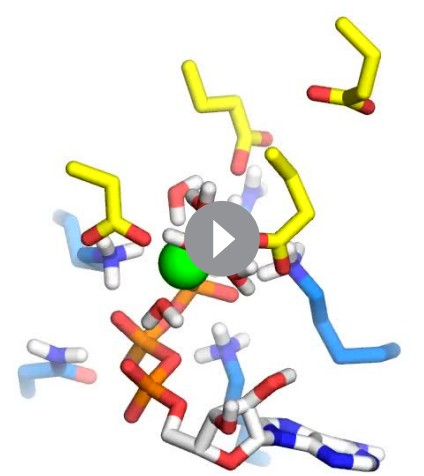

**Video 1.** Close-up of the binding site for ATP-$Mg^{2+}$ in P2X3, in the ATP-bound state, in a representative simulation snapshot (same as *Figure 2E* and *Figure 2—figure supplement 2E*). Residue numbers are indicated in *Figure 2—figure supplement 3*.

DOI: https://doi.org/10.7554/eLife.47060.008

can be therefore observed within a relatively short simulation time, without imposing a specific direction of movement or time course.

Following this strategy, we produced tens of ATP-dissociation trajectories with varying scaling factors (*Figure 2B*, *Figure 2—figure supplement 1*). We then analyzed our simulation data to determine the most populated configuration of $Mg^{2+}$ once ATP dissociates, and how it differs from that preferred in the ATP-bound state. In the presence of ATP and $Mg^{2+}$, the simulation data (obtained with a conventional algorithm) is highly consistent with our crystal structures, showing stable close-range interactions between $Mg^{2+}$ and both the ATP γ-phosphate and D158 (*Figure 2C*, *Figure 2—figure supplement 2*). The ion retains part of its first hydration shell, and water molecules mediate additional interactions with E156, E111 and E109 (*Figure 2—figure supplement 2*, *Video 1*). The γ- and β-phosphates are also tightly coordinated by multiple basic sidechains in the receptor, some of which concurrently form salt-bridges with the acidic sidechains coordinating $Mg^{2+}$, that is K299-D158 and R281-E156 (*Figure 2—figure supplement 2*, *Video 1*). Altogether, this network of electrostatic interactions appears to mediate a highly stable complex with a well-defined geometry. Following ATP dissociation, this mode of $Mg^{2+}$ binding changes noticeably and reproducibly. We detect a clear upward shift in the preferred location of $Mg^{2+}$ (*Figure 2C*), consistent for all three protein subunits (*Figure 2D*), despite the fact that ATP dissociation from each site occurs at different time points in the simulations. This displacement is seemingly driven by a re-optimization of ion-protein interactions upon ATP release. Specifically, D158, E109 and E111 reorient (*Figure 2E,F*) to interact with the ion at a closer distance, whether directly or via water molecules, while E156 is approximately unchanged (so are the K299-D158, R281-E156 salt-bridges) (*Figure 2—figure supplement 2*, *Video 2*). Altogether, these results suggest the existence of a pathway whereby $Mg^{2+}$ bound to P2X3 receptors exchanges dynamically between the two distinct modes.

## Divalent cations stabilize ATP on hP2X3 receptor channels

Dissociation of ATP plays a key role in the function of P2X3 receptors as this process is required for recovery from desensitization. The observation that divalent ions mediate additional interactions between ATP and P2X3 suggests that the ions might slow ATP dissociation. We examined this possibility computationally by inducing ATP-dissociation trajectories from a receptor lacking a divalent ion bound, following the same protocol outlined above. Using this data and that obtained for $Mg^{2+}$-ATP bound P2X3, we derived the apparent ATP dissociation probability as a function of the interaction-scaling factor, in the absence or presence of $Mg^{2+}$. As anticipated, the simulation results show that the ATP dissociation probability increases as the value of the scaling factor decreases, whether $Mg^{2+}$ is present or not, with sigmoidal-like dependence (*Figure 2G*). Crucially, though, the presence of $Mg^{2+}$ produces a clear left-shift in this trend, indicating that $Mg^{2+}$ strengthens the effective ATP-receptor interaction. Moreover, the sigmoidal slope becomes steeper (by about a factor of 2), indicating $Mg^{2+}$ and receptor act cooperatively. Indeed, in many simulated trajectories we observe $Mg^{2+}$ holding on to a partially unbound nucleotide, fostering re-binding (*Figure 2—figure supplement 1*, *Figure 2—figure supplement 3*). Altogether these results substantiate the hypothesis that the presence of $Mg^{2+}$ in our ATP-bound P2X3 structures is not incidental; rather, the ion appears to be a dominant contributing factor to the lifetime and geometry of the ligand-receptor complex.

To experimentally test this hypothesis, we used electrophysiological measurements. Characterizing the dissociation of ATP using functional approaches is not possible for wild-type hP2X3 because the receptor desensitizes much

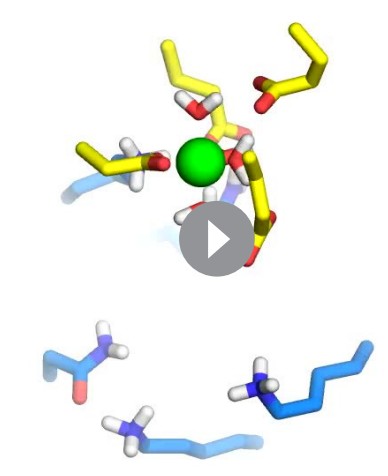

**Video 2.** Close-up of the binding site for ATP-$Mg^{2+}$ in P2X3, in the ATP-dissociated state, in a representative simulation snapshot (same as *Figure 2F* and *Figure 2—figure supplement 2F*). Residue numbers are indicated in *Figure 2—figure supplement 2F*.
DOI: https://doi.org/10.7554/eLife.47060.009

faster than ATP unbinds (*Cook and McCleskey, 1997*; *Giniatullin and Nistri, 2013*; *Sokolova et al., 2006*). Instead we used whole-cell patch clamp recordings of a slowly desensitizing mutant, and measured the time required for channel closure (deactivation) after removing external ATP, both in the absence and presence of divalent cations. ATP unbinding is the rate-limiting step for closure or deactivation of the slowly desensitizing P2X receptor channels (*Rettinger and Schmalzing, 2004*), and we therefore used the kinetics of channel deactivation in hP2X3 slow as a measure of the kinetics of ATP unbinding. The hP2X3 slow construct used for these experiments was generated by mutating three residues in the N-terminus (T13P/S15V/V16I) to greatly slow desensitization (*Figure 3A*), as previously described (*Hausmann et al., 2014*; *Mansoor et al., 2016*). In the absence of divalent cations, the hP2X3 slow construct deactivates with a time constant of $2.1 \pm 0.3$ s following removal of external ATP (*Figure 3B,C*). By contrast, the presence of 1 mM $Mg^{2+}$ slowed deactivation by ~8-fold, while 1 mM $Ca^{2+}$ slowed deactivation by ~ 4 fold (*Figure 3B,C*), suggesting that both ions slow ATP dissociation. In addition to these results obtained at pH 7.3, we also tested the effect of $Mg^{2+}$ at pH 5.0, the acidic condition used for crystallization, and observed that 1 mM $Mg^{2+}$ increased the deactivation time constant from $0.7 \pm 0.1$ s to $6.6 \pm 0.6$ s (*Figure 3C*, *Figure 3—figure supplement 1*). The deactivation time constants were faster at pH 5.0, in agreement with previous reports that P2X3 receptors have a lower affinity for ATP under acidic conditions (*Gerevich et al., 2007*; *Wildman et al., 1999*). Although both $Mg^{2+}$ and $Ca^{2+}$ ions slow hP2X3 slow deactivation, the effect of $Ca^{2+}$ is more modest when compared to $Mg^{2+}$, suggesting that $Ca^{2+}$ might compete with $Mg^{2+}$ to hasten unbinding under physiological conditions. We estimated divalent cation affinity by measuring the deactivation time constant in the presence of different concentrations of $Mg^{2+}$ or $Ca^{2+}$, and obtained an $EC_{50}$ of 102 μM for $Mg^{2+}$ and 39 μM for $Ca^{2+}$ (*Figure 3D,E*), revealing that all receptors are regulated by divalent ions under physiological conditions and that $Ca^{2+}$ might play a more prominent role. When we tested a physiological solution containing 1 mM $Mg^{2+}$ and 2 mM $Ca^{2+}$, we measured a deactivation time constant of $10.2 \pm 1.8$ s, close to that with 1 mM $Ca^{2+}$ (*Figure 3B,C*), even though the concentrations of $Ca^{2+}$-ATP and $Mg^{2+}$-ATP would be similar in this solution because the stability constant for $Ca^{2+}$ binding to ATP is weaker than for $Mg^{2+}$. In addition to prolonging deactivation, stabilizing ATP on the receptor would be expected to increase the apparent affinity for ATP activation of the receptor. We therefore constructed concentration-response relations for ATP activation of hP2X3 slow in the absence and presence of $Mg^{2+}$. We found that 0.5 mM $Mg^{2+}$ shifts the $EC_{50}$ from 40 nM to 18 nM, demonstrating that $Mg^{2+}$ increases the apparent ATP affinity for the receptor (*Figure 3—figure supplement 2*).

## The lower mode of $Mg^{2+}$ binding is essential for stabilizing ATP on the receptor

The structures of $Mg^{2+}$-ATP and $Ca^{2+}$-ATP bound P2X3 receptors suggest that D158 is a critical residue for the lower mode of divalent cation coordination. When D158 is mutated to Ala in hP2X3 slow, we observed considerably slower deactivation in the absence of divalent ions ($\tau = 7.3 \pm 0.4$ s) compared to the control construct, demonstrating that mutations near the ATP binding site can influence dissociation of ATP. Remarkably, instead of slowing deactivation, $Mg^{2+}$ actually accelerated deactivation of the D158A mutant, with an affinity somewhat weaker than observed for $Mg^{2+}$ slowing of deactivation for the control receptor (*Figure 3F,G*). We also studied the E156A mutation in hP2X3 slow and found that the mutation did not appreciably alter deactivation in the absence of divalent ions (*Figure 3—figure supplement 3A*), but $Mg^{2+}$ accelerated channel deactivation, with an affinity that is also lower than the control receptor (*Figure 3—figure supplement 3B*), suggesting that $Mg^{2+}$ can still bind to hP2X3 with lower affinity when either E156 or D158 is mutated, and that both E156 and D158 are required for $Mg^{2+}$ to stabilize ATP on the receptor. In contrast, when the crucial residue E109 involved in the upper mode of coordination is mutated to Ala, $Mg^{2+}$ slowed deactivation as in the control receptor (*Figure 3—figure supplement 3C,D*), suggesting that this residue is not involved in stabilizing ATP. When all three residues E109-E156-D158 are mutated to Ala in hP2X3 slow, $Mg^{2+}$ has no discernable effect on channel deactivation (*Figure 3H,I*), revealing that acidic residues contributing to the lower mode of divalent binding are essential for the ions to interact with and stabilize ATP.

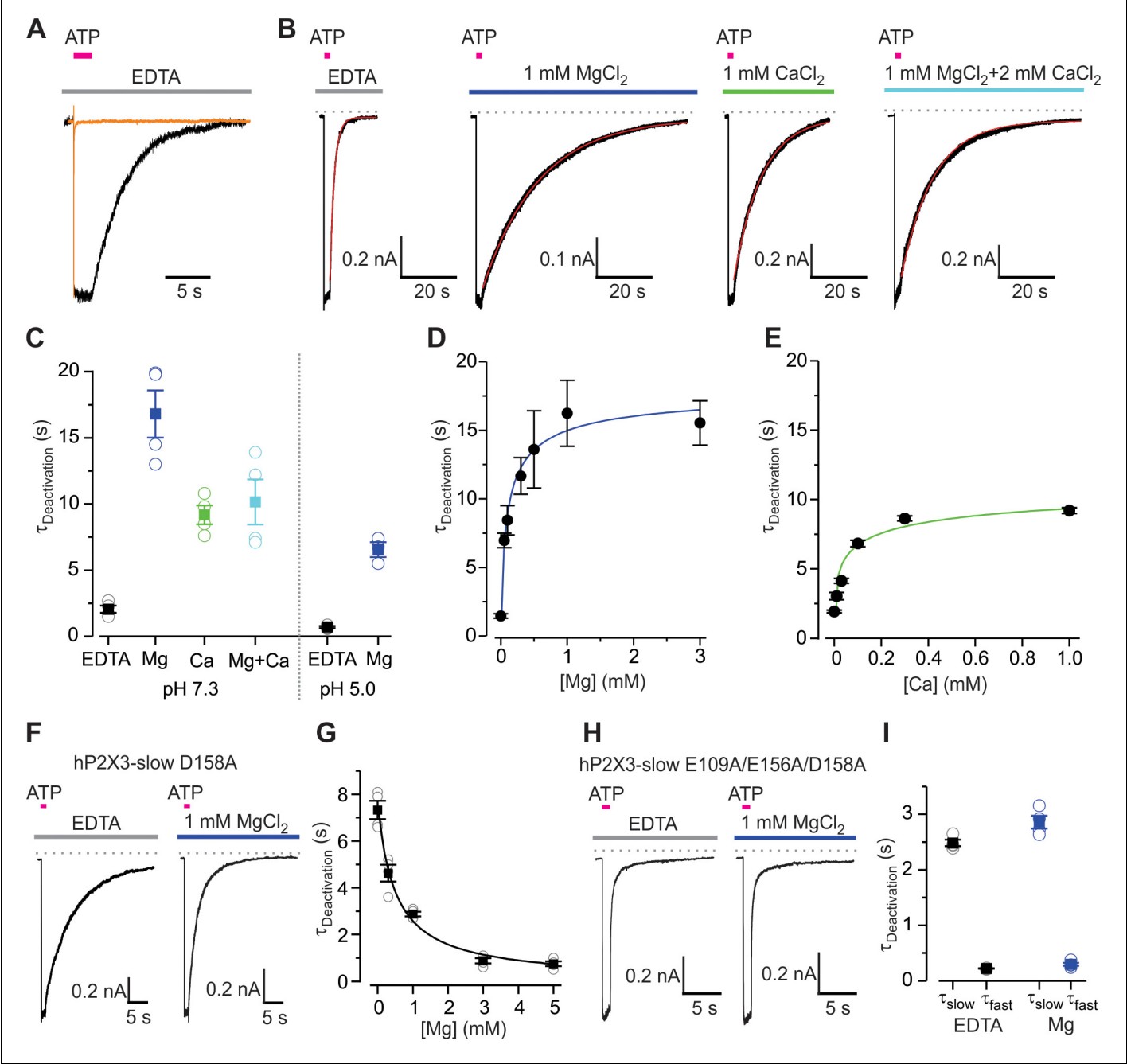

**Figure 3.** $Mg^{2+}$ and $Ca^{2+}$ slow deactivation of hP2X3 receptors. (**A**) Superimposed fast-desensitizing hP2X3 wt (orange) and slow-desensitizing hP2X3 slow (black) macroscopic receptor channel currents activated by extracellular application of a saturating concentration of free ATP (10 µM) in the presence of EDTA (10 mM). (**B**) hP2X3 slow receptor channel currents activated by ATP (10 µM) were recorded in different extracellular divalent cation concentrations at pH 7.3 and shown on the same time scale. A single exponential function was fit to current deactivation and fits are shown in red, superimposed with the currents in black. (**C**) Summary of deactivation time constants in the absence and presence of different divalent cations at neutral or acidic pH, with individual cells shown in open symbols and filled symbols representing the mean and error bars S.E.M. (n = 4 for each). (**D and E**) Concentration-response relations for divalent cation slowing hP2X3 slow channel deactivation at pH 7.3. Each data point is the mean ± S.E.M. with n = 3. The Hill equation was fit to the data in D ($Mg^{2+}$) with $EC_{50}$ = 102 ± 56 µM and $n_H$ = 0.7 ± 0.3, and to the data in E ($Ca^{2+}$) with $EC_{50}$ = 39 ± 6 µM and $n_H$ = 0.5 ± 0.4. (**F**) 100 µM ATP activated macroscopic currents for D158A hP2X3 slow were recorded in the absence or presence of $MgCl_2$. (**G**) Deactivation time constants obtained from single exponential fits to current decay recorded after removing ATP in the absence and presence of $Mg^{2+}$ at different concentrations. Filled symbols are the mean ± S.E.M. with n = 4, individual measurements are shown as open symbols. The Hill equation was fit to the data ($EC_{50}$ = 482 ± 106 µM, $n_H$ = 0.9). (**H**) 300 µM ATP activated macroscopic currents for the E109A/E156A/D158A hP2X3 slow triple

*Figure 3 continued on next page*

*Figure 3 continued*

mutant were recorded in the absence or presence of MgCl₂. (I) Deactivation time constants obtained from double exponential fits to current deactivation. Filled symbols are the mean ± S.E.M. with n = 4, individual measurements are shown as open symbols.

DOI: https://doi.org/10.7554/eLife.47060.010

The following figure supplements are available for figure 3:

**Figure supplement 1.** $Mg^{2+}$ slows deactivation of hP2X3 slow receptor channel currents at acid pH.

DOI: https://doi.org/10.7554/eLife.47060.011

**Figure supplement 2.** $Mg^{2+}$ increases the apparent affinity for activation of hP2X3 slow receptor channels by ATP.

DOI: https://doi.org/10.7554/eLife.47060.012

**Figure supplement 3.** Effects of $Mg^{2+}$ on single mutations in the lower or upper modes within the divalent cation binding chamber.

DOI: https://doi.org/10.7554/eLife.47060.013

## Divalent cations slow hP2X3 recovery from desensitization

The wild-type hP2X3 receptor desensitizes much more rapidly than ATP unbinds, leading to a very slow recovery from desensitization. Since $Mg^{2+}$ binding to the lower mode slows ATP dissociation, we hypothesized that the reported inhibitory effects of $Mg^{2+}$ (*Giniatullin et al., 2003*; *Li et al., 2013*) might result from the ion stabilizing a desensitized state (*Figure 4—figure supplement 1A*). Moreover, we suspected the reported facilitation of P2X3 receptor by $Ca^{2+}$ observed in solutions containing $Mg^{2+}$ (*Cook et al., 1998*) might be related to our observation that $Ca^{2+}$ slows unbinding of ATP less dramatically than $Mg^{2+}$ and that the influence of $Ca^{2+}$ prevails when both divalent ions are present at physiological concentrations (*Figure 3B–E*). To investigate how divalent cations influence gating of the wild-type hP2X3 receptor, we used a rapid piezo-controlled solution exchange system to apply saturating concentration of ATP for 500 ms to activate channels in the absence or presence of either $Mg^{2+}$ or $Ca^{2+}$ and examined the resulting current kinetics. In all cases, hP2X3 receptor channels activate rapidly and currents reach a maximum value within 1 ms (*Figure 4A*), a value that is close to the limit of our solution exchange. Once activated, channels rapidly desensitize in the continuous presence of ATP, with the kinetics of current decay being well fit by a double exponential function, with time constants (τ) that were not discernably different in the absence or presence of either $Mg^{2+}$, $Ca^{2+}$ or a physiological concentration of the two ions (*Figure 4A,B*). We then investigated the kinetics of recovery from desensitization by reapplying ATP at different time points after the initial ATP application that activates and desensitizes the receptors. The time constant for recovery from desensitization in the absence of divalent cations was 63 ± 8 s (*Figure 4C*), significantly faster than previous results obtained using physiological concentrations of divalent ions (*Cook and McCleskey, 1997*; *Cook et al., 1998*). In contrast, recovery from desensitization was slowed by almost 4-fold (τ = 248 ± 71 s) in the presence of $Mg^{2+}$ (1 mM) and to a lesser extent by external $Ca^{2+}$ (τ = 115 ± 42 s at 1 mM) (*Figure 4C*). We also examined the kinetics of recovery from desensitization in the presence of physiological concentrations of $Mg^{2+}$ and $Ca^{2+}$ and obtained a time constant of 140 ± 44 s, close to that observed with 1 mM $Ca^{2+}$ (*Figure 4C*). These results demonstrate that $Mg^{2+}$ (and to a lesser extent $Ca^{2+}$) stabilizes a desensitized state of the hP2X3 receptor, most likely by increasing stability of the receptor-ATP complex.

To further link the effects of divalent ions on ATP dissociation and recovery from desensitization, we examined the effects of the D158A mutant. Interestingly, although the onset of desensitization for D158A is similar to wild-type (*Figure 4D,E*), the mutant slowed recovery from desensitization in the absence of divalent ions, consistent with slower deactivation of D158A mutant in hP2X3 slow (*Figure 4F*) (τ = 140 s for D158A compared to τ = 63 s for wild-type). $Mg^{2+}$ hastened recovery from desensitization in D158A, an effect that is similar to the acceleration of ATP dissociation by $Mg^{2+}$ in hP2X3 slow D158A (*Figure 4F*). We also examined the D109A/D156A/D158A triple mutant, and observed that the mutant exhibited slower desensitization, faster recovery from desensitization and that $Mg^{2+}$ had no discernable effect on either of these measures (*Figure 4G–I*). From these results we conclude that $Mg^{2+}$ binding to the lower mode observed in our structures stabilizes ATP on the receptor and slows recovery from desensitization. Stabilization of a desensitized state by $Mg^{2+}$ can explain why this ion inhibits the activity of P2X3 receptor channels when the nucleotide is applied at a frequency that is sufficient for recovery from desensitization in the absence of the divalent, but not in its presence (*Giniatullin et al., 2003*; *Li et al., 2013*) (*Figure 4—figure supplement 1A,B*).

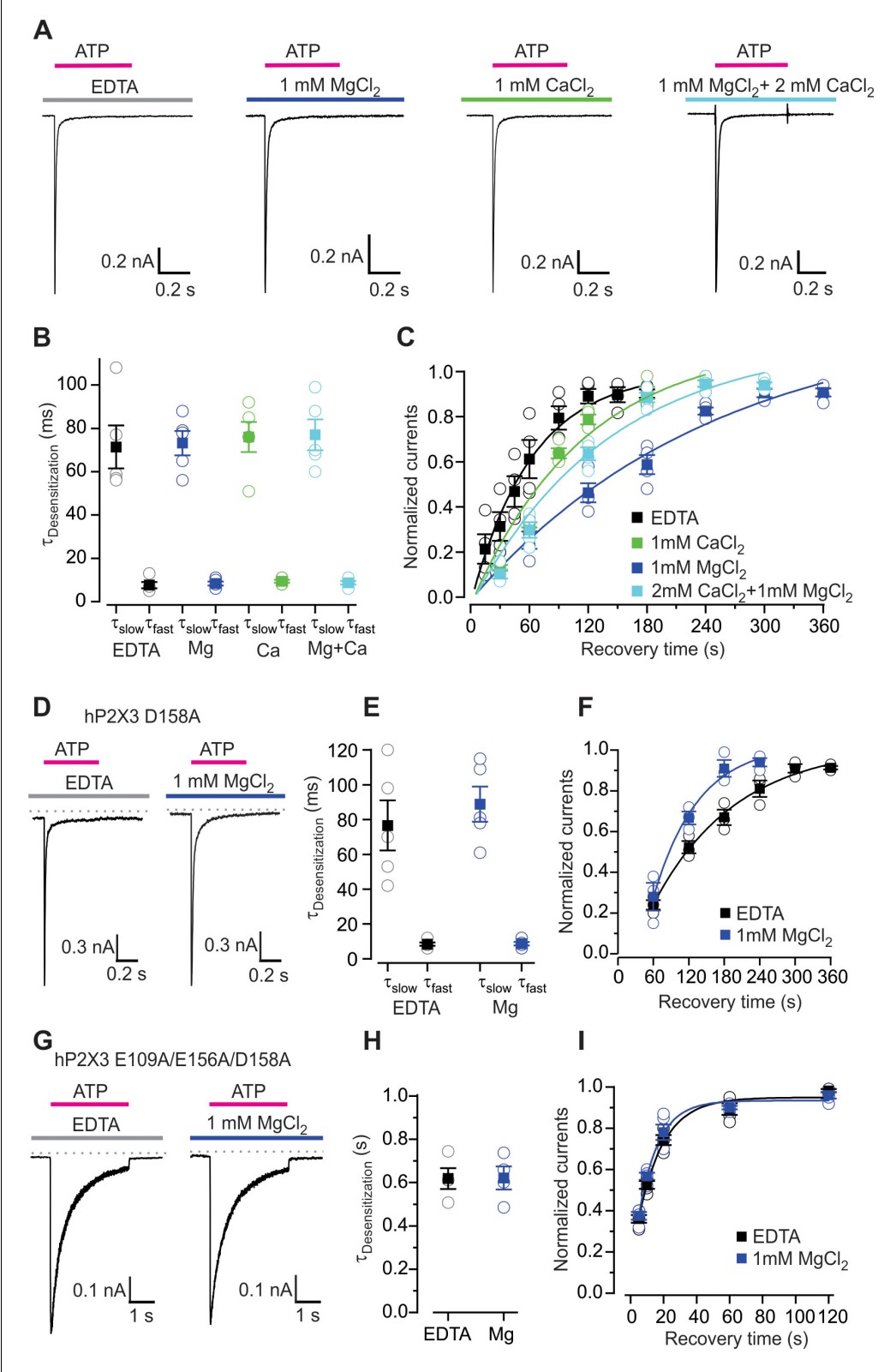

**Figure 4.** Influence of Mg$^{2+}$ and Ca$^{2+}$ on desensitization of hP2X3 receptor channels. (**A**) Macroscopic wt hP2X3 receptor channel currents activated by a saturating concentration of ATP (300 μM) under different conditions. (**B**) Summary of desensitization time constants obtained by double exponential fits of desensitizing currents in the absence and presence of Mg$^{2+}$ or Ca$^{2+}$, with individual cells shown in open symbols and mean values in filled symbols, with error bars representing S.E.M. (n = 5 for each). (**C**) Time courses for recovery from desensitization of hP2X3 receptor channels in the

*Figure 4 continued on next page*

*Figure 4 continued*

absence and presence of divalent cations. Responses were normalized to the amplitude of current activated by the first application of a saturating concentration of ATP (300 μM) for 500 ms (n = 4). Smooth lines are single exponential fits to the time courses with τ = 63 ± 8 s for EDTA, τ = 248 ± 71 s for Mg$^{2+}$, τ = 115 ± 42 s for Ca$^{2+}$ and τ = 140 ± 44 s for Mg$^{2+}$ plus Ca$^{2+}$. (D) 300 μM ATP activated macroscopic currents for D158A were recorded in the absence or presence of MgCl$_2$. (E) Desensitization time constants obtained by fitting a double exponential function to the current decay during ATP (100 μM) application with individual cells shown in open symbols and mean values in filled symbols, with error bars representing S.E.M. (n = 5 for each). (F) Time courses for recovery from desensitization obtained as in panel C (n = 3). Smooth lines are single exponential fits to the time courses with τ = 140 ± 21 s for EDTA and τ = 76 ± 25 s for Mg$^{2+}$. (G) ATP activated macroscopic currents recorded for the E109A/E156A/D158A triple mutant were recorded in response to 1 mM ATP in the absence or presence of MgCl$_2$. (H) Desensitization time constants obtained by single exponential fits to the current decay during ATP (1 mM) application with individual cells shown in open symbols and mean values in filled symbols, with error bars representing S.E.M. (n = 4 for each). (I) Time courses for recovery from desensitization obtained as in panel C (n = 5). Smooth lines are single exponential fits to the time courses with τ = 15 ± 3 s for EDTA and τ = 12 ± 2 s for Mg$^{2+}$.

DOI: https://doi.org/10.7554/eLife.47060.014

The following figure supplement is available for figure 4:

**Figure supplement 1| A.** Triple mutation in the Mg$^{2+}$ binding chamber eliminates inhibition caused by Mg$^{2+}$.

DOI: https://doi.org/10.7554/eLife.47060.015

## The divalent cation binding chamber is crucial for activation by Mg$^{2+}$-ATP

We next explored the role of the two modes of divalent binding for activation of hP2X3 receptor channels by Mg$^{2+}$-ATP using a protocol where ATP is applied along with enough Mg$^{2+}$ for Mg$^{2+}$-ATP to be the major form of the nucleotide present in solution (*Figure 5*). Using this protocol, the wild type hP2X3 as well as hP2X3 slow receptors can be robustly activated by a solution containing a low concentration of free ATP and a high concentration of Mg$^{2+}$-ATP (*Li et al., 2013*) (*Figure 5A–C*, *Figure 5—figure supplement 1*). For the D158A and E156A mutants that contribute to the lower mode of divalent binding, although the extent of activation by Mg$^{2+}$-ATP was somewhat less than for the wild-type receptor, we still observed robust activation by Mg$^{2+}$-ATP (*Figure 5A–C*). For the E109A and E111A mutants within the upper mode of divalent binding, activation by Mg$^{2+}$-ATP was indistinguishable from the wild type receptor (*Figure 5—figure supplement 2A–C*). In contrast, when both E156A and D158A mutants were combined, we observed considerably weakened activation by Mg$^{2+}$-ATP (*Figure 5A–C*), indicating that the lower mode is important for Mg$^{2+}$-ATP activation. We also observed a 'resurgent' current upon removal of Mg$^{2+}$-ATP from the external solution in the E156A/D158A double mutant (*Figure 5B*). A similar resurgent current was previously observed for the P2X2 receptor following removal of Mg$^{2+}$-ATP solutions, where we established that the receptor can bind both free ATP and Mg$^{2+}$-ATP, but that only free ATP can robustly promote channel opening (*Li et al., 2013*). In the case of P2X2 receptors, the resurgent current appears following removal of Mg$^{2+}$-ATP because Mg$^{2+}$ unbinds more rapidly than free ATP, giving rise to the transient activation by free ATP remaining on the receptor (*Figure 5—figure supplement 3B,C*). Interestingly, when either the D158A or E156A mutants in the lower mode were combined with either the E109A or E111A mutants within the upper mode, we also observed greatly reduced activation by Mg$^{2+}$-ATP and a resurgent current upon removal of Mg$^{2+}$-ATP (*Figure 5—figure supplement 2A–C*). For the triple mutant (E09A/E156A/D158A) involving both lower and upper modes of divalent ion binding, we saw no discernable activation by Mg$^{2+}$-ATP in either hP2X3 or hP2X3 slow, but Mg$^{2+}$-ATP concentration dependent resurgent current (*Figure 5—figure supplement 1*, *Figure 5—figure supplement 2*, *Figure 5—figure supplement 3A–C*). From these findings we conclude that the lower mode of divalent binding to P2X3 receptors is essential for activation by divalent bound ATP and that the upper mode also plays a detectable role.

## Discussion

The goal of the present study was to understand how P2X3 receptors are activated by ATP in the presence of physiological concentrations of Ca$^{2+}$ and Mg$^{2+}$ where most of the nucleotide is divalent-bound. Collectively, our results show how the receptor uses an acidic chamber to accommodate those ions in two distinct modes; either bound in the lower mode where the ion interacts with the γ-phosphate of ATP, or in the upper mode in the absence of the nucleotide. The interaction between

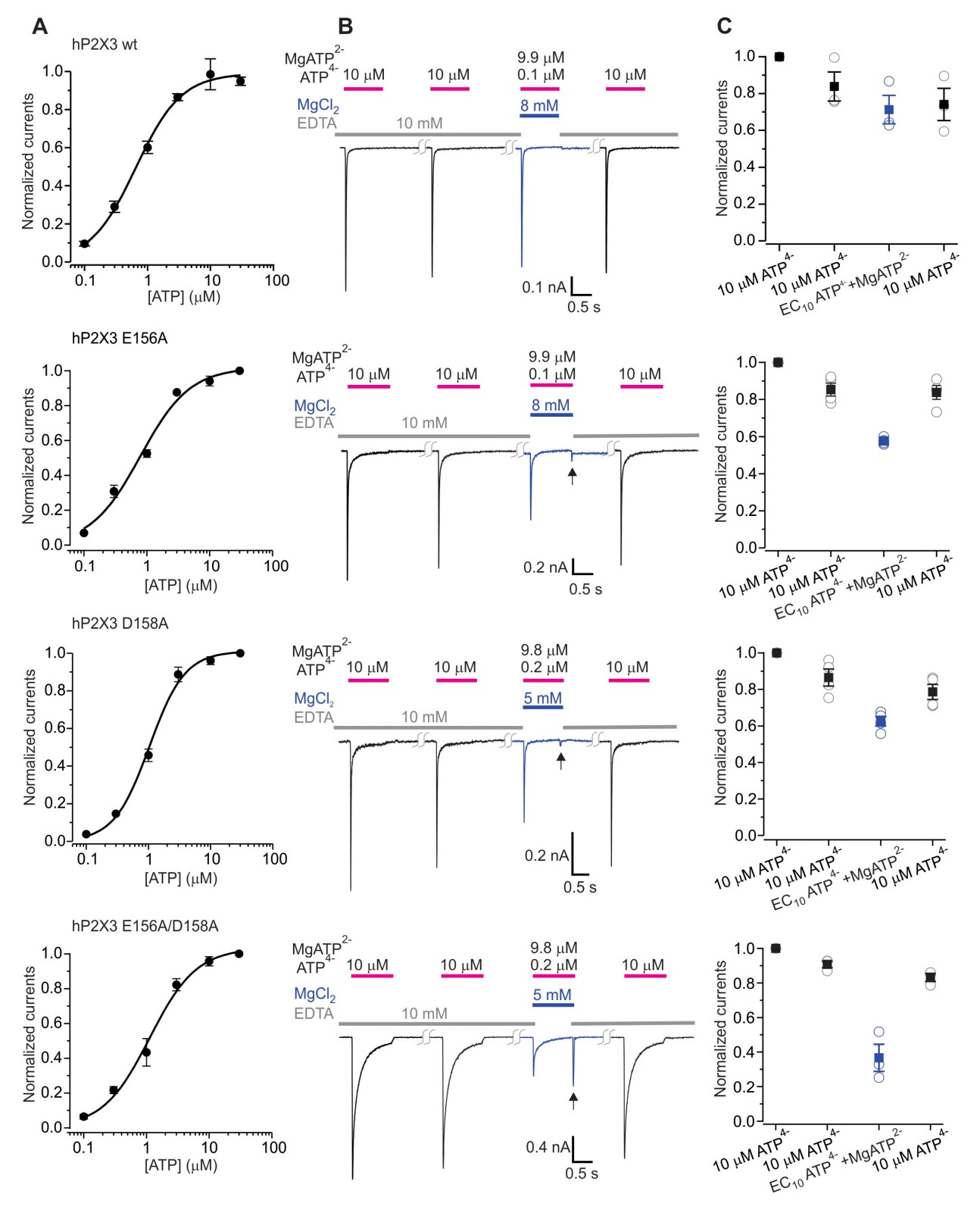

**Figure 5.** The divalent cation binding chamber is critical for Mg$^{2+}$-ATP activation. (**A**) Concentration-response relations for ATP activation of wt hP2X3, E156A, D158A and the E156A/D158A double mutant in divalent-free solution. Each data point is the mean of four measurements (± S.E.M.). Smooth lines are fits of the Hill equation to the data where EC$_{50}$ and $n_H$ are 0.7 ± 0.1 μM and 1.2 ± 0.1 for wt hP2X3, 0.8 ± 0.1 μM and 1.1 ± 0.2 for E156A, 1.0 ± 0.1 μM and 1.5 ± 0.2 for D158A and 1.1 ± 0.2 μM and 1.2 ± 0.2 for E156A/D158A. (**B**) Macroscopic currents activated with solutions containing

*Figure 5 continued on next page*

*Figure 5 continued*

either a saturating concentration of free ATP or one containing mostly $Mg^{2+}$-ATP plus an $EC_{10}$ quantity of free ATP for wt hP2X3, E156A, D158A or the E156A/D158 double mutant. The first applications of free ATP at 2 min intervals were used to estimate the extent of rundown between consecutive applications of ATP. Note the resurgent current (black arrows) for E156A, D158A and the E156A/D158A double mutant upon removal of $Mg^{2+}$-ATP solution. (C) Summary of currents in response to each ATP application, normalized to the amplitude of the first. Open symbols are individual cells and fill symbols are the mean, with error bars representing S.E.M. (n = 3 for wt and the double mutant and n = 4 for all others).

DOI: https://doi.org/10.7554/eLife.47060.016

The following figure supplements are available for figure 5:

**Figure supplement 1.** The divalent cation binding chamber is critical for Mg-ATP activation of hP2X3 slow.

DOI: https://doi.org/10.7554/eLife.47060.017

**Figure supplement 2.** The divalent cation binding chamber is critical for Mg-ATP activation of hP2X3.

DOI: https://doi.org/10.7554/eLife.47060.018

**Figure supplement 3.** $Mg^{2+}$-ATP can bind but not activate hP2X3 slow receptors when the acidic chamber is mutated.

DOI: https://doi.org/10.7554/eLife.47060.019

divalent ions and the ATP γ-phosphate seen in our crystal structures is distinct from that in most other structures in the PDB and also from that typical of ATP in solution; in both cases, the divalent ions preferentially interact with both β-and γ-phosphates of the nucleotide. Our computational and functional results collectively show that binding of $Mg^{2+}$ stabilizes ATP on the receptor, slowing unbinding and recovery from desensitization, and increasing the apparent affinity for ATP, with similar results for $Ca^{2+}$ in functional experiments. Our mutagenesis results show that the acidic chamber is required for activation of the receptor when divalent-bound forms of ATP predominate in solution. Although the lower mode of divalent binding is critical for activation by divalent-bound ATP, the upper mode also seems to play a role because combining mutations in the two modes diminishes activation when divalent-bound forms of the nucleotide predominate.

Our results suggest that the following sequence of events occur as divalent-bound ATP binds to activate P2X3 receptors (*Figure 6*). We know that $Mg^{2+}$ can bind to the upper mode seen in the X-ray structure of apo hP2X3 when present in mM concentrations (*Mansoor et al., 2016*), so we assume that the upper mode would be occupied by $Mg^{2+}$ (and likely $Ca^{2+}$) under physiological conditions prior to the recognition of divalent-bound ATP. In one scenario, initial engagement of divalent-bound ATP with the receptor would weaken the interaction of the ion bound to the β- and γ-phosphates of ATP; in addition, dissociation of the ion would be fostered by the ion occupying the upper mode in the acidic chamber, as this ion shifts to the lower mode to interact with the γ-phosphate of ATP, while still engaged to E156 and D158. In our simulations, both ATP and $Mg^{2+}$ (in the lower mode) remain stably associated with the receptor, and when ATP dissociation is accelerated the ion spontaneously moves from the lower mode to the upper mode, demonstrating the existence of a pathway for the movement of divalent ions between the two modes and reinforcing the idea that ion binding within the upper mode is favored in the absence of the nucleotide. An alternate scenario would be that when divalent-bound ATP is recognized by the receptor, it is the divalent ion originally occupying the upper mode that dissociates, allowing the ion bound to β− and γ−phosphates of ATP to reposition to the lower mode seen in our X-ray structures. Our simulations do not permit us to discriminate between these two scenarios, as they probe the dissociation of ATP only. Nevertheless, two arguments can be put forward in favor of the first scenario. From a structural standpoint, dissociation of the ion occupying the upper mode appears less likely because it would be sterically hindered not only by the surrounding receptor but also ATP itself; by contrast, the ion bound to ATP would be completely exposed to the solvent. In addition, a mechanism whereby the ion bound to ATP must dissociate to permit complete agonist recognition and channel activation would be applicable to P2X2 receptors and P2X3 E109A/E156A/D158A mutants, which as noted above, can bind divalent-bound ATP even though the complex does not promote robust activation of the receptor.

In both of these scenarios, the P2X3 receptor uses the acidic chamber to enable activation and to stabilize ATP on the receptor in the presence of divalent ions. Following receptor activation and desensitization, dissociation of ATP would be much slower in the presence of divalent ions, in no small part due to the divalent ion anchoring the nucleotide to D158. Not only does this ion (at this position) strengthen the effective ATP-receptor interaction; it also fosters re-binding of ATP

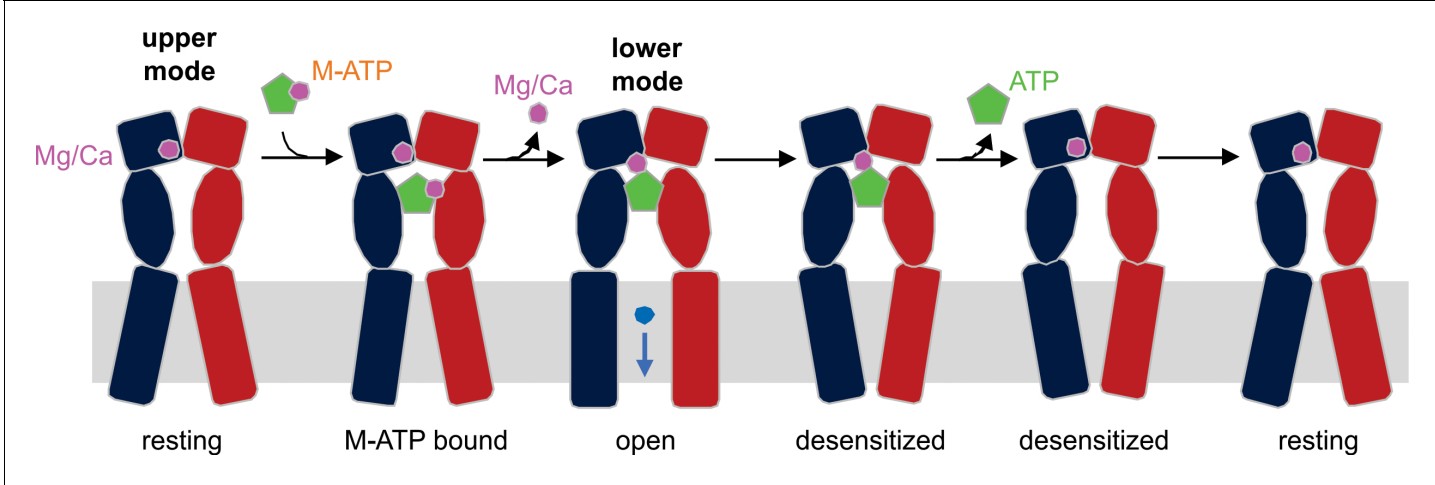

**Figure 6.** Schematic illustration of the mechanism of $Mg^{2+}$-ATP/$Ca^{2+}$-ATP activation of P2X3 receptor channels. $Mg^{2+}$ or $Ca^{2+}$ binds to hP2X3 receptors in the upper mode under physiological conditions. When ATP divalent cation complex (M-ATP) enters the nucleotide binding pocket, a divalent cation will rapidly dissociate, leaving one to be coordinated by the γ-phosphate of ATP and residues E156 and D158. The divalent cation sandwiched between ATP and receptor strengthens the effective ATP-receptor interaction, and also fosters re-binding of ATP molecules that become transiently dislodged from the nucleotide-binding pocket, rendering a very slow recovery from desensitization for hP2X3 receptors. Only two of three subunits of a hP2X3 receptor are shown for clarity.

DOI: https://doi.org/10.7554/eLife.47060.020

molecules that become transiently dislodged from the nucleotide-binding pocket, that is the dissociation (and recognition) process becomes cooperative. When the wild-type P2X3 receptor is not occupied by $Ca^{2+}$ or $Mg^{2+}$, and ATP is recognized in its free form under divalent-free laboratory conditions, binding is likely suboptimal because D158 is electrostatically repulsive with the γ-phosphate of ATP. Consistent with this hypothesis, when D158 is mutated to Ala, receptors actually hold on to ATP more strongly than the wild-type in the absence of divalent ions, as reflected in slower ATP dissociation and recovery from desensitization (*Figure 3C,G*; *Figure 4C,F*). In contrast, for the D158A mutant, when the agonist is an ATP-divalent complex, the incoming ion bound to the nucleotide would be less likely to move away from β− and γ−phosphates of ATP, and ATP would thus spend more time in a conformation that is in part incompatible with the intricate network of receptor-ligand-solvent interactions observed in the wild-type receptor, resulting in accelerated ATP dissociation and recovery from desensitization (*Figure 3G*; *Figure 4F*). Consistent with this model, a priori disruption of this network, for example in the E109A/E156A/D158A mutant, results in minimal $Mg^{2+}$ effects, but ATP binding is markedly weakened, with dramatically accelerated dissociation and recovery from desensitization (*Figure 3I*; *Figure 4I*).

In conclusion, we have uncovered a previously unrecognized structural element in P2X3 receptors that is essential for channel activation under physiological conditions where divalent-bound forms of ATP predominate. Our results also reveal that divalent ions play a fundamental role in tuning the gating properties of P2X3 receptors. By hampering ATP dissociation, acting cooperatively with the protein, divalent ions slow down the receptor recovery from desensitization, thereby reducing the frequency with which the channel is available for activation by extracellular ATP. Interestingly, the influence of $Ca^{2+}$ on stabilizing ATP on the receptor and slowing recovery from desensitization is less pronounced than for $Mg^{2+}$, which could explain why $Ca^{2+}$ can facilitate activation of P2X3 receptors in physiological solutions containing $Mg^{2+}$ in response to repeated applications of ATP (*Cook and McCleskey, 1997*; *Cook et al., 1998*). Appreciating the pronounced influence of divalent ions on the gating properties of P2X3 receptors will be essential for understanding how these receptors are involved in sensory processes in the nervous system.

## Materials and methods

### Purification, crystallization and structure determination

The crystallization construct of human P2X3 ΔN5ΔC33-T13P/S15V/V16I (hP2X3 MFC$_{slow}$) were expressed and purified as described previously (*Wang et al., 2018*). For the crystallization of hP2X3 MFC$_{slow}$ with ATP and Mg$^{2+}$, 1 µl of hP2X3 MFC$_{slow}$ at 2–3 mg/ml and 1 mM ATP were mixed with 1 µl of the reservoir solution (0.05 M Mg-acetate, 0.05 M Na-acetate pH 5.0 and 21.2–25.8% PEG400). For the crystallization with ATP and Ca$^{2+}$, 1 µl of hP2X3 MFC$_{slow}$ at 2–3 mg/ml and 1 mM ATP were mixed with 1 µl of the reservoir solution (0.2 M NaCl, 0.05 M Ca-acetate pH 5.0 and 21.2–25.8% PEG 400). Both crystals with Mg$^{2+}$ and Ca$^{2+}$ ions grew using the vapor diffusion method over the reservoir resolutions after 3 weeks, and were harvested with the respective cryoprotectant solutions for the crystals with Mg$^{2+}$ (0.05 M Mg-acetate, 0.05 M Na-acetate pH 5.0, 40% PEG400 and 1 mM ATP) and for the crystals with Ca$^{2+}$ (0.2 M NaCl, 0.05 M Ca-acetate pH 5.0, 40% PEG 400 and 1 mM ATP) before cryo-cooling. All X-ray diffraction data sets were collected at the SPring-8 beamline BL41XU. The data sets were processed using XDS2 (*Kabsch, 2010*). Both ATP and Mg$^{2+}$-bound and ATP and Ca$^{2+}$-bound hP2X3 structures were solved by molecular replacement with Phaser, using the ATP-bound, open state structure of hP2X3 (PDB ID: 5SVK) as a template. The models were further improved by iterative cycles of manual modeling with COOT3 (*Emsley and Cowtan, 2004*). and refined using PHENIX4 (*Adams et al., 2010*). Data collection and refinement statistics are shown in *Supplementary file 1*.

### Survey of ATP-Mg$^{2+}$ complexes in the Protein Data Bank

The statistical analysis described in *Figure 2A* comprises entries in the PDB obtained through X-ray crystallography at a resolution higher than or equal to 2.5 Å and containing at least one ATP bound to one Mg$^{2+}$ (or Ca$^{2+}$) ion. Redundant structures with more that 70% of sequence identity were eliminated from this set. Structures in which ATP is bound concurrently to Mg$^{2+}$(or Ca$^{2+}$) and other cations were also eliminated. The resulting sets includes 171 entries (12 entries for Ca$^{2+}$). An analogous set of 152 entries was obtained for Mg$^{2+}$ complexes with the non-hydrolysable ATP analog AMP-PNP. Both sets were divided into classes according the proximity of Mg$^{2+}$ and each of the phosphate groups in the ATP tri-phosphate moiety. For example in the configuration referred as 'βγ', Mg$^{2+}$ is directly and concurrently coordinated only by oxygen atoms in the β- and γ-phosphate groups. To carry out this classification in practice, we defined and calculated the following 'ion-coordination number' for each of the phosphate groups and each of the PDB entries:

$$N^s = \frac{R}{\log \sum_i \exp[R/d(\mathrm{O}_i^s, \mathrm{Mg}^{2+})]}$$

where $d\,(\mathrm{O}, \mathrm{Mg}^{2+})$ denotes an oxygen-Mg$^{2+}$ distance, $s$ denotes either the α-, β- or γ-phosphates, and $i$ denotes each of the oxygen atoms in each phosphate not forming the ATP backbone (i.e. three atoms in γ, and two atoms in α and β). For $R = 1{,}000$ Å, a value of $N^s$ smaller than 3.2 Å reflects direct coordination by the $s$-phosphate; for example in the αβγ geometry $N^{\alpha}$, $N^{\beta}$ and $N^{\gamma}$ are concurrently smaller than this threshold.

### Molecular dynamics simulations

All MD simulations were carried out with NAMD version 2.13 (*Phillips et al., 2005*), using the CHARMM36 force field (*Best et al., 2012*), at constant temperature (298 K) and pressure (1 atm). The integration time-step was 2 fs. Electrostatic interactions were calculated using the PME method with a real-space cut-off of 12 Å; van-der-Waals interactions were also cut-off at 12 Å, with a switching function turned on at 10 Å. Periodic boundary conditions were used. The simulations of ATP-bound P2X3 in the absence of Mg$^{2+}$ were based on PDB entry 5SVK (*Mansoor et al., 2016*). Those of P2X3 bound to ATP and Mg$^{2+}$ were based on PDB entry 5SVL (*Mansoor et al., 2016*). In the latter, a density signal near the ATP γ-phosphate designated as Na$^+$ in the PDB entry was reassigned as Mg$^{2+}$; this assignment is confirmed by the crystal structures reported here. Each of the simulation systems is a cubic box of side approximately equal to 100 Å. The systems comprise the extracellular domain of P2X3 (residues 47 to 317), immersed in a 100 mM KCl solution (plus counterions neutralizing the total charge). Disulfide bonds were assumed between residues 107 and 153, 116 and 137,

122 and 147, 203 and 213, and 247 and 256, in all three protomers. Following a PROPKA evaluation (*Bas et al., 2008*), all ionizable residues were set to be in their default state at pH 7, with the exception of Glu109 in the $Mg^{2+}$-free ATP-bound structure. Three types of simulations were carried out for each system, in three consecutive stages. The first stage, or equilibration, comprised a sequence of simulations during which the dynamics of the protein-ligand complex is limited, primarily through internal-coordinate restraints (RMSD, distance restraints), to a degree that is gradually diminished over a period of 85 ns. In the second stage, a simulation of 50 ns was carried whereby the structural dynamics of the complex was for the most part unrestricted; however, to preclude large deviations resulting from the deletion of the transmembrane domain, a weak restraint was applied to approximately maintain the relative arrangement of the secondary-structure elements of the protein as in the experimental structure of the full-length complex. Specifically, this restraint is a harmonic function of the RMSD of the backbone, relative to the X-ray structure, with force constant $k = 1$ kcal/mol/Å². On average, the RMSD values of the resulting ensembles of protein configurations are $1.5 \pm 0.2$ Å for the secondary-structure core and $2.3 \pm 0.3$ Å for all backbone; that is this weak restraint is such that the variability of the configurations explored by the protein is not unlike what could be expected for a simulation of the full-length protein. A third set of MD simulations were carried out subsequently for both the P2X3-ATP and P2X3-ATP-$Mg^{2+}$ complex, whereby the dissociation of ATP was induced using the so-called solute-tempering method (*Jo and Jiang, 2015*; *Liu et al., 2005*). That is, the energy function was modified so as to scale down the non-bonded interactions of ATP with all other components in the simulation system by a factor $S = \sqrt{\lambda}$; non-bonded interactions within ATP were also scaled, down, by a factor $\lambda$. After a series of initial tests, a total of 80 independent simulations of 50 ns each were carried out for the P2X3-ATP complex, for $\lambda$ values equal to 0.20, 0.25, 0.30 and 0.35 (20 trajectories for each value of $\lambda$). An Analogously for the P2X3-ATP-$Mg^{2+}$ complex, 120 simulations were carried out with $\lambda$ values equal to 0.12, 0.15, 0.175, 0.20, 0.25 and 0.30. The dissociation of ATP from each of the 3 protein subunits was monitored over time by evaluating (1) the distance between the centers-of-mass of ATP and the residues defining its binding site (Thr172, Lys63, Lys65, Arg281, Arg299; hydrogen atoms excluded), or $D_1$, and (2) the distance between the ATP γ-phosphorus atom and the $Mg^{2+}$ ion, or $D_2$. In our simulation of the P2X3-ATP-$Mg^{2+}$ complex with unperturbed intramolecular interactions, the characteristic values of $D_1$ and $D_2$ are $5.9 \pm 0.3$ and $3.2 \pm 0.1$ Å, respectively. The density maps, snapshots and distributions shown in *Figure 2C-D, 2E*, *Figure 2—figure supplement 1A-C, E* and *Video 1* derive from this simulation. To evaluate the ATP-dissociated state, from the solute-tempering trajectories, we first considered all snapshots for which $D_1 > 12$ Å and concurrently $D_2 > 5$ Å (each binding site in the protein trimer was analyzed separately). The distributions shown in *Figure 2A-D* describe these configurations, combining the data obtained for all $\lambda$ values and all binding sites. (Note that the solute-tempering scheme does not perturb the inter- or intra-molecular interactions for the receptor, $Mg^{2+}$ and solvent, so once ATP dissociates, the dynamics of the P2X3-$Mg^{2+}$ complex is represented with the standard energy function.) These distributions are however not stationary, as they reflect an evolving configuration of the P2X3-$Mg^{2+}$ complex. For clarity, therefore, the density maps and snapshots shown in *Figure 2C-D, 2F*, *Figure 2—figure supplement 2F* and *Video 2* reflect the subset of this data that appears representative of the emerging most-populated configuration, namely all snapshots of the P2X3-$Mg^{2+}$ complex (with ATP dissociated) in which the distance between $Mg^{2+}$ and the carboxyl groups of D158, E156, E111 and E109 is equal or smaller than 6 Å.

## Cell culture and transfection for electrophysiology

Human embryonic kidney 293 (HEK-293) cells were purchased from ATCC (ATCC CRL-1573), cultured in Dulbecco's Modified Eagle's Medium (DMEM) supplemented with 10% fetal bovine serum and 10 mg/L gentamicin. Following an initial passage, cells were frozen in liquid nitrogen using Freezing Media (Life Technologies) and these aliquots used to generate subsequent cultures. To minimize cell variability, cells were used with fewer than 15 passages (<2 months). All cell culture reagents were obtained from Life Technologies. Trypsin treated HEK293 cells were seeded onto glass coverslips in 35 mm dish or 6-well plates the day of transfection and placed in a 37°C incubator with 95% air and 5% $CO_2$. Transfections were performed using FuGENE6 Transfection Reagent (Promega). hP2X3 receptors were co-transfected with a green fluorescent protein (GFP) cDNA construct pcDNA3-Clover (addgene) at a ratio of 2:1. Cells were used for whole-cell recording 18–30 hr

after transfection. hP2X3 in pCMV6-XL4 was purchased from OriGENE and then cloned into pcDNA3. All the mutations were introduced into hP2X3 using QuikChange II site-directed mutagenesis kit and verified by DNA sequencing.

## Whole-cell patch clamp recording

Whole-cell patch clamp recordings were used to record P2X receptor channel currents from transiently transfected HEK-293 cells. Membrane currents were recorded under voltage-clamp using an Axopatch 200B patch clamp amplifier (Axon Instruments, Inc) and digitized on-line using a Digidata 1322A interface board and pCLAMP 9.2 software (Axon Instruments, Inc). Currents were filtered at 2 kHz using 8-pole Bessel filters, and digitized at 5 or 10 kHz. The pipette solution for standard whole-cell recording on HEK cells contained (mM): 140 NaCl, 10 EGTA and 10 HEPES, adjusted to pH 7.0 with NaOH. The divalent-free extracellular solution contained (mM): 140 NaCl, 10 EDTA, and 10 HEPES, adjusted to pH 7.3 or 5.0 with NaOH. Assuming that divalent ion contaminations from reagents and water is less than 1 µM, any possible free divalent cations in the 10 mM EDTA solution would be less than 0.5 nM. The extracellular solutions with different concentrations of $Mg^{2+}$ or $Ca^{2+}$ contained (mM): 140 NaCl, 10 HEPES, and desired concentration of $MgCl_2$ or $CaCl_2$, adjusted to pH 7.3 or 5.0. $Na_2ATP$ (referred to as ATP) was used in all the experiments. The stability constants of ATP binding to cations present in our solutions are: $Mg^{2+}$ $10^{4.10}$, $Ca^{2+}$ $10^{3.76}$, $Na^+$ $10^{0.83}$, and $K^+$ $10^{1.17}$ (NIST Critically Selected Stability Constants of Metal Complexes). Max Chelator (http://max-chelator.stanford.edu) was used to calculate concentrations of $Mg^{2+}$, $MgATP^{2-}$ and $ATP^{4-}$ in solutions with different concentrations of $MgCl_2$ and ATP. Recording solutions were adjusted to the correct pH after adding $MgCl_2$ and ATP. Solution exchange was achieved using either the Rapid Solution Changer RSC-200 (BioLogic), which can change solution within ~ 50 ms and has the capacity of switching between nine solutions to construct dose-response relationships, or a Piezo driven system (AutoMate Scientific), which can change between two solutions within 1 ms to resolve rapidly desensitizing currents. To generate concentration-response relationships, a reference concentration of ATP was applied before each test concentration was applied, as previously described (*Li et al., 2008*). The Hill equation was fit to the data according to: $I/I_{max} = [ATP]^{nH} / ([ATP]^{nH} + EC_{50}^{nH})$, where I is the normalized current at a given concentration of ATP, $I_{max}$ is the maximum normalized current, $EC_{50}$ is the concentration of ATP ([ATP]) producing half-maximal currents and $n_H$ is the slope factor.

## Acknowledgements

We thank Mark Mayer, Miguel Holmgren and members of the Swartz, Faraldo-Gómez and Hattori laboratories for helpful discussions. We also thank Giacomo Fiorin for assistance with simulation and data analysis tools. This work utilized the computational resources of the NIH HPC Biowulf cluster (http://hpc.nih.gov). We thank the staff from BL41XU beamline at SPring-8 (Proposal Nos. 2017B2523 and 2018A2507), from BL19U1 beamline of National Facility for Protein Science Shanghai (NFPS) at Shanghai Synchrotron Radiation Facility (SSRF) (Proposal Nos. 2016-NFPS-PT-001047 and 2017-NFPS-PT-001191), and from BL17U1 at SSRF (Proposal Nos. 15ssrf02687 and 2016-SSRF-PT-005911), for assistance during data collection.

## Additional information

### Competing interests

Kenton Jon Swartz: Reviewing editor, *eLife*. José D Faraldo-Gómez: Reviewing editor, *eLife*. The other authors declare that no competing interests exist.

### Funding

| Funder | Author |
| --- | --- |
| National Institute of Neurological Disorders and Stroke | Kenton Jon Swartz |
| National Heart, Lung, and Blood Institute | José D Faraldo-Gómez |

| Key Technologies Research and Development Program | Motoyuki Hattori |
| National Natural Science Foundation of China | Motoyuki Hattori |
| State Key Laboratory of Genetic Engineering | Motoyuki Hattori |
| Office of Global Experts Recruitment in China | Motoyuki Hattori |

The funders had no role in study design, data collection and interpretation, or the decision to submit the work for publication.

## Author contributions

Mufeng Li, Conceptualization, Data curation, Formal analysis, Validation, Investigation, Visualization, Writing—original draft; Yao Wang, Rahul Banerjee, Fabrizio Marinelli, Data curation, Formal analysis, Validation, Investigation; Shai Silberberg, Conceptualization, Investigation; José D Faraldo-Gómez, Conceptualization, Formal Analysis, Investigation, Visualization, Supervision, Funding acquisition, Writing—original draft, Writing—review and editing; Motoyuki Hattori, Conceptualization, Data curation, Formal analysis, Supervision, Funding acquisition, Validation, Investigation, Visualization, Writing—original draft, Writing—review and editing; Kenton Jon Swartz, Conceptualization, Supervision, Funding acquisition, Investigation, Writing—original draft, Project administration, Writing—review and editing

## Author ORCIDs

Fabrizio Marinelli http://orcid.org/0000-0003-0044-6718
José D Faraldo-Gómez https://orcid.org/0000-0001-7224-7676
Motoyuki Hattori http://orcid.org/0000-0002-5327-5337
Kenton Jon Swartz https://orcid.org/0000-0003-3419-0765

## Decision letter and Author response

Decision letter https://doi.org/10.7554/eLife.47060.028
Author response https://doi.org/10.7554/eLife.47060.029

# Additional files

## Supplementary files

• Supplementary file 1. Data collection, phasing and refinement statistics.
DOI: https://doi.org/10.7554/eLife.47060.021
• Transparent reporting form
DOI: https://doi.org/10.7554/eLife.47060.022

## Data availability

All data needed to evaluate the conclusions in this paper are available in the main text and supplementary materials. Refined atomic models have been deposited with the Protein Data Bank (entry codes 6AH4 and 6AH5).

The following datasets were generated:

| Author(s) | Year | Dataset title | Dataset URL | Database and Identifier |
| --- | --- | --- | --- | --- |
| Hattori M | 2019 | Structure of an ion channel in state I | http://www.rcsb.org/structure/6AH4 | Protein Data Bank, 6AH4 |
| Hattori M | 2019 | Structure of an ion channel in state II | http://www.rcsb.org/structure/6AH5 | Protein Data Bank, 6AH5 |

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
