## [Decision Letter]

Thank you for submitting your article "Molecular mechanisms of P2X3 receptor channel activation and modulation by divalent cation bound ATP" for consideration by *eLife*. Your article has been reviewed by three peer reviewers, including Baron Chanda as the Reviewing Editor and Reviewer #1, and the evaluation has been overseen by Richard Aldrich as the Senior Editor. The following individuals involved in review of your submission have agreed to reveal their identity: Bruce P Bean (Reviewer #2); Lucie Delemotte (Reviewer #3).

The reviewers have discussed the reviews with one another and the Reviewing Editor has drafted this decision to help you prepare a revised submission.

Summary:

P2X3 receptors are involved in sensing nociceptive and gustatory stimuli. They are activated by extracellular ATP and are the targets for inflammatory and neuropathic pain. P2X3 receptors form homotrimeric channels that activate and desensitize within a few milliseconds upon ATP binding. However, these channels recover from desensitization, which is accompanied by ATP unbinding in minutes time scale. This slow recovery from desensitization is necessary to prevent hypersensitivity. In the absence of divalent cations, these channels recover from desensitized state rapidly, which suggests that the divalent cations are essential for normal physiological response. The currently available structures of the ATP bound P2X receptor do not show any evidence of divalent cations. Moreover, the divalent ion present in apo-structure is not compatible with ATP binding. But most importantly, the ATP in physiological solution is complexed to Ca and Mg, whereas the current ATP bound P2X structure implies that ATP is somehow stripped of its divalent cation upon binding to the receptor.

To address these questions, the authors have taken a multipronged approach of combining x-ray crystallography with molecular simulations and functional analyses. Together, they make a compelling case that the tripartite complex formed by a divalent cation, ATP and receptor constitutes the physiologically relevant desensitized/open state structure of the channel. Molecular simulations based on solute tempering method show that the transition from the resting to open state involve a pre-open state where two magnesium ions are bound per subunit. Functional studies reveal that the divalent cations are crucial for slow desensitization and neutralization of acidic residues affect activation by Mg^2+^-ATP. This is a comprehensive and elegant study that addresses a long-standing puzzle in the field and provides new insight into the physiological mechanisms of channel activation.

Essential revisions:

1) Based on my reading of the manuscript, the structure presented in this manuscript represents the channel in the desensitized or open state. I understand that ascribing a functional state to X-ray structure is tricky but it gives non-specialist readers some perspective. Is there a particular reason that the authors are more circumspect?

2) In Figure 3H, the triple mutant E109A/E156A/D158A is activated equally well with and without Mg-ATP but in Figure 5B and C (bottom panel) shows that the double mutant E156A/D158A is not fully activated by Mg-ATP. It is not clear why there is this discrepancy.

3) The subsection title "The divalent cation binding chamber is essential […]". I would replace essential with "crucial" or an equivalent word. The mutated channels are still able to activate albeit not as effectively.

4) In Figure 1, I would like to see an identical view of bound free-ATP as shown in panels B, C and D. The authors are making the case that the free-ATP structure is not compatible with the published receptor structure complexed with Mg but right now that is not obvious. In addition, if the binding site for the free-ATP is different than Mg-ATP than it is a bit surprising that the channels are still able to open in the absence of divalent cations. Is the rest of the structure identical or there are other common features that are involved in mediating the binding signal to the pore gates? Please comment.

5) Subsection “Simulations reveal Mg^2+^ displacement to upper mode results from ATP dissociation”: "Upon ATP recognition, Mg^2+^ relocates to bridge the interaction between agonist and receptor" The simulations have shown that ATP dissociation leads Mg^2+^ to relocate to the upper site. This shows that there is a path for Mg^2+^ to transfer from the site, and the structures show that in the absence of ATP Mg^2+^ sits in the upper site while in the presence of it sits in the lower site in interaction with ATP. However, I don't think the above mentioned statement is absolutely established.

6) Subsection “Divalent cations stabilize ATP on hP2X3 receptor channels”: In the slowly desensitizing mutant, the time required for channel closure is used as a proxy to measure the rate of ATP dissociation. This seems to assume that the unbinding of ATP is rate-limiting in the process that leads to closing. Is that true? Can this be explained explicitly and a citation added?

7) Subsection "The divalent cation binding chamber is essential for activation by Mg^2+^-ATP". I did not understand the origin of the resurgent currents. I think the description of what happens in P2X2 is too brief and the analogy between what is going on here and in these previous experiments should be better explained. Maybe even a scheme would help (Maybe similar to Figure 6 explaining what happens when resurgent currents appear?).

---

## [Author Response]

Essential revisions:1) Based on my reading of the manuscript, the structure presented in this manuscript represents the channel in the desensitized or open state. I understand that ascribing a functional state to X-ray structure is tricky but it gives non-specialist readers some perspective. Is there a particular reason that the authors are more circumspect?

The structures we obtained are similar to the open state structure reported by Mansoor et al., 2016. We now state that the structures are of open-like conformation as previously suggested (Results section paragraph one).

2) In Figure 3H, the triple mutant E109A/E156A/D158A is activated equally well with and without Mg-ATP but in Figure 5B and C (bottom panel) shows that the double mutant E156A/D158A is not fully activated by Mg-ATP. It is not clear why there is this discrepancy.

The two sets of experiments are designed to test different questions and there is no discrepancy. The purpose of the experiments shown in Figure 3H is to test if Mg^2+^ alters the kinetics of channel deactivation, while those in Figure 5 are designed to test whether Mg-ATP can activate the receptor. In Figure 3 a very high concentration (300 μM) of ATP was used to ensure maximal channel activation in the absence and presence of Mg^2+^. In the solution with 300 μM ATP and 1 mM MgCl_2_, there would be about 35 μM free ATP, a near suturing concentration for even the E109A/E156A/D158A triple mutant. In Figure 3H we did not originally include scale bars for current amplitude because we were testing whether Mg^2+^ alters the kinetics of deactivation. However, if one does compare current amplitude, the current activated by 300 μM ATP in EDTA (shown on the left) is ~11% larger than the one activated by 300 μM ATP with 1mM MgCl_2_ (35 μM ATP^4-^+65 μM Mg-ATP, shown on the right). According to the dose-response curve shown in Figure 5—figure supplement 2, we should expect similar current amplitude with 35 μM and 300 μM ATP^4-^. The 11% difference might come from Mg-ATP binding but not activating, competing with free ATP. Indeed, we saw a small resurgent current in some of our experiments with the triple mutant. To clarify that the currents are not of identical size, we have added scale bars for the current amplitude.

The purpose of the experiments shown in Figure 5 is to test whether Mg-ATP can efficiently activate different constructs, so we designed solutions to contain either a saturating concentration of free ATP or an EC10 of free ATP along with a high concentration of Mg-ATP. For example, in the solution with 10 μM ATP and 5 mM MgCl_2_, there would be only about 0.2 μM free ATP, an EC10 concentration for the E156A/D158A double mutant, along with 9.8 μM Mg-ATP.

3) The subsection title "The divalent cation binding chamber is essential [...]". I would replace essential with "crucial" or an equivalent word. The mutated channels are still able to activate albeit not as effectively.

We thank the reviewers for this nuanced suggestion. We have changed the wordings as suggested in the relevant sections.

4) In Figure 1, I would like to see an identical view of bound free-ATP as shown in panels B, C and D. The authors are making the case that the free-ATP structure is not compatible with the published receptor structure complexed with Mg but right now that is not obvious. In addition, if the binding site for the free-ATP is different than Mg-ATP than it is a bit surprising that the channels are still able to open in the absence of divalent cations. Is the rest of the structure identical or there are other common features that are involved in mediating the binding signal to the pore gates? Please comment.

We have revised Figure 1 as suggested to make the comparisons more straightforward. The case we are making, though, is not that the existing structures suggest the binding sites for free-ATP and Mg^2+^-ATP differ; rather, than the position of Mg^2+^ in the Mg^2+^-bound apo structure and of ATP in the ATP-bound structure are not consistent with a direct interaction between Mg^2+^ and ATP in a Mg^2+^-ATP bound state. The new structure indeed shows Mg^2+^ must move to directly interact with the ATP.

The relevant text has been modified as follows:

“When ATP is in solution[…]. Whether these interactions are compatible with those observed between ATP and the receptor in the ATP-bound structure is unclear; if they are not, the presence of divalent ions would effectively weaken the interaction between agonist and receptor, which would be physiologically counterintuitive. It would seem reasonable to anticipate that when bound to the receptor, Mg^2+^ and ATP interact differently than when in solution. Intriguingly […]”

5) Subsection “Simulations reveal Mg^2+^ displacement to upper mode results from ATP dissociation”: "Upon ATP recognition, Mg^2+^ relocates to bridge the interaction between agonist and receptor" The simulations have shown that ATP dissociation leads Mg^2+^ to relocate to the upper site. This shows that there is a path for Mg^2+^ to transfer from the site, and the structures show that in the absence of ATP Mg^2+^ sits in the upper site while in the presence of it sits in the lower site in interaction with ATP. However, I don't think the above mentioned statement is absolutely established.

We agree with the reviewers that there are other possibilities and we have modified this section and have discussed an alternative sequence of events in the Discussion.

We appreciate the reviewers for accurately pointing out that our structures together with simulations have revealed a pathway for divalent cation movement. We have now added a sentence: “Altogether, these results suggest the existence of a pathway whereby Mg^2+^ bound to P2X3 receptors exchanges dynamically between the two distinct modes.”

6) Subsection “Divalent cations stabilize ATP on hP2X3 receptor channels”: In the slowly desensitizing mutant, the time required for channel closure is used as a proxy to measure the rate of ATP dissociation. This seems to assume that the unbinding of ATP is rate-limiting in the process that leads to closing. Is that true? Can this be explained explicitly and a citation added?

We thank the reviewers for this thoughtful suggestion. We have added a sentence and citation to clarify this point. In subsection “Divalent cations stabilize ATP on hP2X3 receptor channels” we have added: “ATP unbinding is the rate-limiting step for closure or deactivation of the slowly desensitizing P2X receptor channels (Rettinger and Schmalzing, 2004), and we therefore used the kinetics of channel deactivation in hP2X3-slow as a measure of the kinetics of ATP unbinding.”

7) Subsection "The divalent cation binding chamber is essential for activation by Mg^2+^-ATP". I did not understand the origin of the resurgent currents. I think the description of what happens in P2X2 is too brief and the analogy between what is going on here and in these previous experiments should be better explained. Maybe even a scheme would help (Maybe similar to Figure 6 explaining what happens when resurgent currents appear?).

We have expanded the description of resurgent currents (subsection “The divalent cation binding chamber is crucial for activation by Mg^2+^-ATP”) and added a cartoon (Figure 5—figure supplement 3) to help the reader understand.